# Pre-Existing Intrarenal Parvovirus B19 Infection May Relate to Antibody-Mediated Rejection in Pediatric Kidney Transplant Patients

**DOI:** 10.3390/ijms24119147

**Published:** 2023-05-23

**Authors:** Nicola Bertazza Partigiani, Susanna Negrisolo, Andrea Carraro, Diana Marzenta, Elisabetta Manaresi, Giorgio Gallinella, Luisa Barzon, Elisa Benetti

**Affiliations:** 1Pediatric Nephrology, Department of Women’s and Children’s Health, University Hospital of Padua, 35128 Padua, Italy; 2Department of Women’s and Children’s Health, University of Padua, 35128 Padua, Italy; 3Laboratory of Immunopathology and Molecular Biology of the Kidney, Department of Women’s and Children’s Health, University of Padova, 35127 Padua, Italy; 4Pediatric Research Institute “IRP Città della Speranza”, 35127 Padua, Italy; 5Department of Pharmacy and Biotechnology, University of Bologna, 40138 Bologna, Italy; 6Department of Molecular Medicine, University of Padua, 35121 Padua, Italy

**Keywords:** Parvovirus B19, ABMR, kidney transplantation, viral infection, humoral rejection

## Abstract

Viral infections can lead to transplant dysfunction, and their possible role in rejection is described. In total, 218 protocol biopsies performed in 106 children at 6, 12 and 24 months after transplantation were analyzed according to Banff ’15. RT-PCR for cytomegalovirus, Epstein-Barr virus, BK virus and Parvovirus B19 was performed on blood and bioptic samples at the time of transplant and each protocol biopsy. The prevalence of intrarenal viral infection increases between 6 and 12 months after transplantation (24% vs. 44%, *p* = 0.007). Intrarenal Parvovirus B19 infection is also associated with antibody-mediated rejection (ABMR) (50% ABMR vs. 19% T-cell-mediated rejection, *p* = 0.04). Moreover, Parvovirus infection is higher at 12 months of follow-up and it decreases at 48 months (40.4% vs. 14%, *p* = 0.02), while in 24% of grafts, Parvovirus is already detectable at the moment of transplantation. Intrarenal Parvovirus B19 infection seems to be related to ABMR in pediatric kidney recipients. The graft itself may be the way of transmission for Parvovirus, so performance of a PCR test for Parvovirus B19 should be considered to identify high-risk patients. Intrarenal Parvovirus infection presents mainly during the first-year post-transplantation; thus, we recommend an active surveillance of donor-specific antibodies (DSA) in patients with intrarenal Parvovirus B19 infection during this period. Indeed, it should be considered a treatment with intravenous immunoglobulins in patients with intrarenal Parvovirus B19 infection and DSA positivity, even in the absence of ABMR criteria for kidney biopsy.

## 1. Introduction

Immunosuppressive therapy greatly contributes to reducing the incidence of acute transplant rejection and improving graft outcomes. However, it can also contribute to spreading viral complications [1,2]. Sensitive molecular methods currently allow the detection of subclinical viral infections and are now routinely adopted in post-transplant viral surveillance protocols [3,4]. Viral surveillance aimed to seek the presence of subclinical infections in order to avoid complications and impairment of allograft outcomes. It is well known that viruses are not only responsible for opportunistic diseases in immunosuppressed patients, such as cytomegalovirus (CMV) pneumonia [5,6], but they can also induce an increase in cytokine production and an altered antigenic expression, which can lead to chronic kidney injury, as demonstrated by BKV nephropathy (BKVAN) and Parvovirus B19 microangiopathy [7,8,9]. Epstein-Barr virus (EBV) and CMV can also cause kidney injury through indirect effects due to the activation of the immune system [10,11]. 

Furthermore, a relationship has also been suggested between opportunistic infections by EBV, CMV and BK virus and rejection, even though it remains unclear whether the reduction of immunosuppressive therapy aimed to control viral infection may contribute to the onset of rejection [12,13,14]. It has been hypothesized that viral infections can lead to allograft dysfunction and acute/chronic allograft rejection, both through direct cytopathic effects and the immune response. 

Parvovirus B19 is a DNA virus with a tropism for the bone marrow and the endothelium. It was described for the first time in kidney transplantation in 1986. Chronic anemia and pure red blood cell aplasia are the most common complications of Parvovirus B19 infection in transplanted patients; however, allograft rejection and dysfunction have also been associated with the infection [15,16,17,18]. The mechanisms of kidney damage are unclear, but they may include direct cytopathic effects on glomerular epithelial cells or endothelial cells and glomerular deposition of immune complexes [17]. Even if intravenous immunoglobulins (IVIG) are commonly used, there is no specific therapy recommended for the treatment of Parvovirus B19 infection [19,20]. A possible role of viruses as a triggering factor for the development of donor-specific anti-HLA antibodies in antibody-mediated rejection (ABMR) has also been observed [21,22]. Considering that there is no specific therapy for most of the viral infections affecting the transplanted kidney and that ABMR represents one of the major risk factors for reduced allograft survival, we evaluated whether local virus-mediated inflammation could be associated with the development of ABMR. The aim of this study was to investigate the relationship between systemic and intrarenal viral infections in kidney recipients and humoral and cellular rejection.

## 2. Results

### 2.1. Population Characteristics

A total of 106 patients were included in the study, 39 females and 67 males. The median age at transplantation was 11 years (5–16 years) and weight was 26.9 kg (15.2–46.9 kg). The population characteristics are listed in Table 1. 

Most transplantations were from non-living donors (71.7%). The most common cause of end-stage kidney disease (ESKD) was congenital anomalies of the kidney and urinary tract (CAKUT) (45%). The second-most frequent cause was genetic diseases (36%), including genetic forms of nephrotic syndrome, Alport syndrome, autosomal recessive, or dominant polycystic kidney disease (ARPKD and ADPKD), cystinosis and type 1-hyperoxaluria. Less frequent conditions were haemolytic uremic syndrome (3%), glomerulopathies (5%), vasculitis (1%) and other specified (3%) or unknown (7%) diseases. 

### 2.2. Histological Analysis

A total of 218 histological samples were analyzed: 153 biopsies were classified as Banff1 (70.3%), 36 as T-cell-mediated rejection (TCMR) (16.5%), 16 as ABMR (7.3%) and 9 as interstitial fibrosis/tubular atrophy (IF/TA) (4.1%), while 4 biopsies presented other diagnoses as BKVAN, C3 glomerulopathy and calcineurin toxicity (1.8%). The distribution of the histological diagnoses at 6, 12 and 24 months after transplantation is listed in Table 2. 

The histological analysis demonstrated an increase in the prevalence of acute and chronic allograft injury (21.7% at 6 months vs. 49.1% at 24 months, *p* = 0.004). Indeed, nearly half of the patients presented with a pathological biopsy at 24 months after transplantation.

### 2.3. Viral Infection

The prevalence of systemic viral infection was 26% and it remained stable over time during the follow-up (26.1% at 6 months, 26.1% at 12 months, 26% at 24 months). The most detected virus was the BK virus, whose prevalence showed a mild and not significant increase at 24 months after transplantation (12% at 6 months vs. 16% at 24 months, *p* = 0.75). EBV prevalence was between 10 and 15%, and it was stable over time. CMV and Parvovirus B19 positivity on blood samples was not frequent in our cohort (prevalence 0–5%).

The prevalence of systemic viral infections was analyzed separately according to the histological findings (Banff1 vs. TCMR–ABMR–IF/TA) and time of follow-up (6, 12, 24 months after transplantation). The results are reported in Table 3. 

The presence of systemic viral infection seemed to be associated with Banff1 at 6 months after transplantation (32% Banff1 vs. 5% TCMR–ABMR–IF/TA, *p* = 0.02), while the remaining analysis did not show differences in terms of prevalence of systemic viral infections with respect to the histological results.

The prevalence of intrarenal viral infection significantly increased between 6 and 12 months after transplantation (24% vs. 44%, *p* = 0.007) and then it remained stable.

Parvovirus B19 was detectable in 23.8% of biopsies, EBV in 13.5%, and BKV in 7.9%, while CMV was isolated only in one biopsy (0.004%).

Intrarenal EBV-DNA detection increased over time (7.6% at 6 months, 12.5% at 12 months, 26% at 24 months, *p* = 0.01), while Parvovirus B19 presented a prevalence of 40.4% at 12 months after transplantation, but a significant decrease at 24 months (14%, *p* = 0.02). Moreover, the PCR viral assay demonstrated the presence of Parvovirus B19 in 24 kidneys at the moment of transplantation (T0 = 23%), while no other viruses were detected.

In addition, the prevalence of intrarenal viral infection was analyzed separately according to the histological findings (Banff1 vs. TCMR–ABMR–IF/TA) and the time of follow-up (6, 12, 24 months after transplantation). Statistical analysis did not show differences in terms of prevalence of intrarenal viral infections with respect to the histological pictures. The results are reported in Table 4.

### 2.4. Correlation between Rejection and Viral Infection

AMBR and TCMR diagnosed biopsies were selected to assess whether the presence of viral infection could be associated with cellular or antibody-mediated rejection. A total of 52 biopsies were included, 36 TCMR and 16 ABMR. The results are reported in Table 5.

The prevalence of systemic viral infection in this cohort was 21.1%, in particular 9.6% BK virus, 15.4% EBV and 0% Parvovirus B19. Otherwise, the prevalence of intrarenal viral infection was 50%, 28.9% for Parvovirus B19, 9.6% for BK virus and 15.4% for EBV. CMV was not detected in either blood or tissue.

The isolation of at least one virus on the tissue was more frequent in subjects with ABMR than in those with TCMR (69% vs. 36%, *p* = 0.034). Furthermore, the intrarenal Parvovirus B19 infection was also significantly more associated with ABMR (50% ABMR vs. 19% TCMR, *p* = 0.04). Kaplan Meier analysis was performed to evaluate the association between intrarenal persistence of Parvovirus B19 and an increased incidence of AMBR (Figure 1), but no statistical significance was reached (*p* > 0.05).

The prevalence of EBV and BK virus in TCMR and AMBR was similar, considering both systemic infection (EBV 13% in ABMR vs. 17% in TCMR, *p* = 1.0; BKV 19% in ABMR vs. 5% in TCMR, *p* = 0.16) and intrarenal infection (EBV 13% in ABMR vs. 17% in TCMR, *p* = 1.0; BKV 13% in ABMR vs. 8% in TCMR, *p* = 0.60).

### 2.5. In Situ Hybridization

In situ hybridization was performed in four histological samples with TCMR in which Parvovirus B19 had been detected through RT-PCR, showing an active viral replication in 25% of the samples. Otherwise, in situ hybridization performed on another four samples with ABMR and a positive RT-PCR for Parvovirus B19 showed an active replication in 75% of samples (Figure 2). However, statistical analysis was not performed due to the small number of samples.

## 3. Discussion

Several pathologic processes, both immune-mediated and not immune-mediated, can reduce graft survival over time, such as acute and chronic rejection and viral infections. Parvovirus B19 is a virus with a tropism for the endothelium that is involved in allograft dysfunction [15,16,17,18]. It is known that Parvovirus B19 infection after renal transplantation can be associated with thrombotic microangiopathy [23]. Our group [18] also related the persistent isolation of intragraft Parvovirus B19 to a greater risk of chronic graft dysfunction according to a former study. In our population, Parvovirus B19 was identified on the tissue in 50% of patients with ABMR, compared to 19% in subjects with TCMR (*p* = 0.04). The possible relationship between the intrarenal Parvovirus B19 infection and ABMR is the most interesting finding of our study, as it has not been previously reported in the literature. Vascular endothelium is the target of Parvovirus B19 [24,25] and it is hypothetical that it could act as an antigen-presenting cell during infections, with an over-exposure of MHC II and the activation of acquired immunity, which would ultimately lead to the development of humoral response and production of donor-specific anti-HLA antibodies, resulting in ABMR [26,27,28]. This hypothesis is supported by the work by Abrahimi et al., who suggested that the risk of acute rejection should be reduced by blocking the expression of class II MHC on the endothelium [29].

An important issue is that Parvovirus B19 should be in active replication to trigger the kidney vascular endothelium. We analyzed a small number of samples with positive Parvovirus B19 DNA on tissue and a histological picture of ABMR or TCRM in order to assess the state of replication of the virus in the endothelial cells. Parvovirus B19 replicated only in 25% of TCMR samples, while it was in active replication in 66% of ABMR samples. Although it was not possible to perform statistical analysis due to the small number of samples analyzed, these preliminary data, not previously reported in the literature, are interesting and worth further confirmation on the largest number of samples. 

According to our results, Parvovirus B19 is present in almost 25% of transplanted kidneys at the time of transplantation, suggesting that the graft could be the source of the transmission of the virus, leading to a primary infection, as already described by Barzon et al. [7]. Yu et al. evaluated donor-derived Parvovirus B19 infection, performing serial serum viral load [30]. The results showed that the incidence of donor-derived Parvovirus B19 infection was 0.4% and 1.5% in living and deceased kidney transplantations, respectively. Parvovirus B19, like other viruses, can remain latent within the organ in immunocompetent subjects, but it can reactivate later in the transplanted patient due to the recipient’s immunosuppression and possible serological mismatch (i.e., Parvovirus IgG positive donor vs. IgG negative recipient), especially in small children, who are often non-immune to Parvovirus B19 infection, at the time of transplantation. 

In our cohort, the prevalence of intrarenal Parvovirus B19 infection increased during the first year post-transplant and then decreased in the second year (40.4 vs. 14% *p* = 0.002), maybe because the tapering of immunosuppression in the second year after transplantation allows a better immune response and the clearance of local infection.

Considering both the relationship between intrarenal Parvovirus B19 and ABMR and the possible transmission of Parvovirus B19 by graft, we suggest Parvovirus B19-DNA analysis on a small tissue sample collected at the time of transplantation. Then, we advise that patients with intrarenal Parvovirus B19 detection should perform active surveillance of donor-specific antibodies (DSA), especially during the first year after transplantation. This could represent an effective strategy for ABMR prevention.

Although guidelines from the American Society of Transplantation Infectious Diseases confirm that no proven specific preventive strategy against Parvovirus B19 infection is available, IVIG has been reported to be beneficial in recipients with Parvovirus B19 infection [31,32]. However, the optimal dosing regimen and duration of IVIG therapy for Parvovirus B19 infection have not been established, and some patients have been reported to have long-lasting resolution of the infection even without IVIG therapy [19]. IVIG is also included in ABMR treatment protocols [30]. Therefore, treatment with IVIG in patients with intragraft Parvovirus B19 detection and DSA onset during surveillance, even in the absence of ABMR criteria for kidney biopsy, should be considered as a long-term preventive strategy for ABMR.

As regards the other viruses analyzed in our study, no further associations between systemic or intrarenal viral detection and acute or chronic rejection were identified (Table 3 and Table 4). 

CMV infection was rare in our cohort (only 2 intrarenal isolations of CMV-DNA and 4 positive CMV-DNA in blood in the months preceding the biopsy). Even though it is known that CMV can trigger an immune response, which can result in chronic allograft injury and acute rejection [12], the application of solid protocols of prophylaxis with effective antiviral drugs in high-risk patients probably reduces the incidence of acute CMV infections [33]. 

Our data did not highlight a relationship between EBV and rejection. Indeed, the possible relationship between EBV infection and rejection remains uncertain in the literature. Some studies have suggested that active EBV infection may be associated with acute rejection in patients suffering from late graft injury [14,34]. Puliyanda et al. recently presented a case series about the efficacy of rituximab in PTLD, finding that patients treated with rituximab for persistent positive EBV-DNA in blood seemed to present a lower incidence of development of de novo DSA (dnDSA) and ABMR, suggesting a role of EBV in AMBR [21]. However, in our cohort, the prevalence of intrarenal EBV infection increased over time during follow-up (7.6% at 6 months, 12.5% at 12 months, 26% at 24 months, *p* = 0.01), even if the prevalence of systemic infection remained stable over time, but this outcome can be affected by the short follow-up time. 

BK virus did not seem to be related to acute or chronic rejection (5% TCMR, 19% ABMR, *p* = 0.16) in our population. Instead, the BKV virus has a recognized role in the development of de novo DSA (dnDSA) and ABMR in the literature. Sawinski et al. suggested that persistent BK viremia is associated with an increased risk for de novo DSA and other studies confirmed this suspicion [13,22,35,36]. The mechanism is still unknown, even if the reduction of immunosuppressive therapy may contribute [37]. In a recent study, Cheungpasitporn observed that BKVAN, but not BK virus infection, was a risk factor for dnDSA and subsequent ABMR [38]. A possible reason for disagreement with our results and literature could be the retrospective nature of the study (DSA was not available for all patients with BKV positivity) and the small number of biopsies with acute or chronic rejection, which could have affected the statistical analysis. 

Considering the overall viral infections, our results showed that a systemic viral infection seems to be associated with Banff1 at 6 months after transplantation (Banff1 32% vs. rejection/IFTA 5% *p* = 0.02). There are no data in the literature, but this is probably a bias due to the strong immunosuppression during the first 6 months after transplantation, which can favor primary or secondary viral infections in recipients. Furthermore, the incidence of rejection or kidney damage is lower at 6 months (21.7%), so it is probable that the concomitance of these conditions is linked to a wrong association.

## 4. Materials and Methods

### 4.1. Patients

This is a retrospective study, which included children who underwent kidney transplantation at our Centre between January 2011 and July 2017 and whose protocol biopsies, performed 6, 12 and 24 months after transplantation, were adequate for histological analysis according to the Banff 2015 criteria [39]. For each patient, age, weight, sex, condition leading to end-stage kidney disease (ESKD), transplant number, source of donation (living/not living), donor/recipient (D/R) weight ratio, number of HLA mismatch, HLA sensitization (defined as panel-reactive antibody, PRA > 80%) and renal function expressed as creatinine clearance (calculated according to Bedside Swartz’s formula) were recorded [40]. According to our local protocol, the immunosuppressive regimen included basiliximab or anti-thymocyte globulin induction, calcineurin inhibitor, mofetil mycophenolate and tapered doses of prednisone [41,42,43]. All patients received anti–human cytomegalovirus (CMV) prophylaxis for 6 months, except for recipients who were serologically negative for CMV and whose donors were negative [44]. All patients received trimethoprim-sulfamethoxazole antibiotic prophylaxis for 6 months after transplantation [45]. Acute T-cell rejections were treated with three methylprednisolone pulses (500 mg/m^2^/dose). Acute ABMRs were treated with three methylprednisolone pulses (500 mg/m^2^/dose), five sessions of plasma exchange, five administrations of intravenous immunoglobulins (200 mg/kg/dose) and one rituximab infusion (375 mg/m^2^) [46,47].

### 4.2. Histological Data

Protocol allograft biopsies were performed 6, 12 and 24 months after transplantation. Two tissue samples were obtained with ultrasound guidance using a Tru-cut 16 G needle and stained with hematoxylin and eosin, Masson’s trichrome, PAS and silver PAS. C4d and CD68 staining was analyzed by means of immunohistochemical techniques. For the purpose of the study, all biopsy specimens were blindly reviewed by an expert pediatric nephrologist and classified according to the Banff 2015 classification as normal (Banff I), acute or chronic ABMR (Banff II), acute or chronic T-cell-mediated rejection (TCMR) (Banff IV) and interstitial fibrosis with tubular atrophy (IFTA) (Banff V). Biopsy specimens showing BK virus-related nephropathy (BKVAN) or relapse of underlying disease (i.e., vasculitis, C3 glomerulopathy) were excluded from the study.

### 4.3. Virological Data

Viral surveillance was performed as described elsewhere [18]. Briefly, EBV, CMV, BK virus and B19 DNAemia were measured by quantitative real-time polymerase chain reaction (RT-PCR) on blood samples. IgM and IgG antibody testing for HCMV infection was performed by ELISA using a commercially available kit (Enzygnost; Dade Behring, Germany). Testing for IgM antibodies against EBV–viral capsid antigen (VCA) and IgG antibodies against EBV–early antigen (EA), EBV-VCA, and EBV–Epstein-Barr nuclear antigen (EBNA) was performed by ELISA (DiaSorin; Saluggia, Italy). Parvovirus B19 IgM and IgG were detected by enzyme immunoassay (Biotrin; Dublin, Ireland). Quantitative real-time PCR was performed on the very day of transplantation (baseline) and then weekly until day 30, every 2 weeks until month 3, monthly until month 12, and then every 3 months until month 24. Additional testing was performed on patients with clinical signs or symptoms of infection. Serological assay was performed at baseline and then at 6, 12 and 24 months or in case of detection of viral DNAemia. Subclinical viral infection was defined as the positivity of viral DNAemia (positivity of Parvovirus B19-DNA and >1000 copies/ml of EBV-, CMV-, and BK virus-DNA) for at least 3 of the 6 months before the biopsy, in the absence of symptoms. Symptomatic viral infection was defined as the detection of viral DNAemia in a patient with acute symptoms and eventual seroconversion after the infection.

### 4.4. Detection of Viral Nucleic Acids in Allograft Biopsy Specimens

All biopsy specimens were screened for CMV-, EBV-, BK virus- and Parvovirus B19- DNA positivity using quantitative RT-PCR. DNA was purified from frozen tissue sections using the QIAamp DNA Mini Kit (Qiagen; Hilden, Germany). Approximately 100 ng of total DNA was analyzed for quantitative RT-PCR with the oligonucleotide primers and TaqMan probes reported elsewhere [18]. The detection limit of RT-PCR methods utilized was 0.01 copies/1000 cells and it was determined by serial dilutions of a plasmid containing the PCR fragment in a sample negative for the presence of viruses. A few samples were also analyzed using in situ hybridization (ISH) with a standard protocol: a DNA probe of genomic length was labeled with digoxigenin and then detected with anti-digoxigenin antibody conjugated with alkaline phosphatase, with colorimetric development with NBT/BCIP [48]. Three different observers evaluated the results, and the slides were repeated if there was doubt. In the few cases that remained uncertain (no consensus among the three observers), a cautious interpretation (negative rather than false positive) was preferred.

### 4.5. Donor-Specific Antibodies (DSA)

The presence of anti-HLA antibodies was tested on blood samples by means of Luminex^®^ method and classified as donor-specific (DSA) or non-DSA (NDSA). DSA > 3000 MFI were considered clinically significant [49]. Anti-HLA antibodies were tested on histological indications (histological picture of suspect ABMR) before January 2016 and then at the time of protocol biopsies after January 2016. 

### 4.6. Statistical Analysis

Continuous quantitative variables with non-normal distribution and discrete quantitative variables were expressed as median and interquartile ranges. Categorical variables were expressed as an absolute value or a percentage. The X2 test or Fisher’s exact test were applied to compare qualitative variables between two groups, depending on the appropriateness of the test, and Kaplan Meier analysis was performed in case of statistical significance of the previous tests. A *p*-value < 0.05 was considered statistically significant.

## 5. Conclusions

In conclusion, intrarenal Parvovirus B19 infection seems to be related to ABMR in pediatric kidney recipients. The graft itself may be the source of transmission of Parvovirus, so an RT-PCR test for Parvovirus B19-DNA at the time of transplantation should be considered to identify high-risk patients and to undertake active surveillance of DSA in patients with intrarenal Parvovirus B19 detection, especially during the first year after transplantation. Active interventions, such as IVIG course, in patients with intrarenal Parvovirus B19 infection and DSA positivity should also be considered, even in the absence of ABMR criteria for kidney biopsy. Even though our results should be confirmed by a longitudinal study and multivariate analysis on a larger population, they also suggest that in situ hybridization should be implemented for the detection of active replicating viruses in the graft.

## Figures and Tables

**Figure 1 ijms-24-09147-f001:**
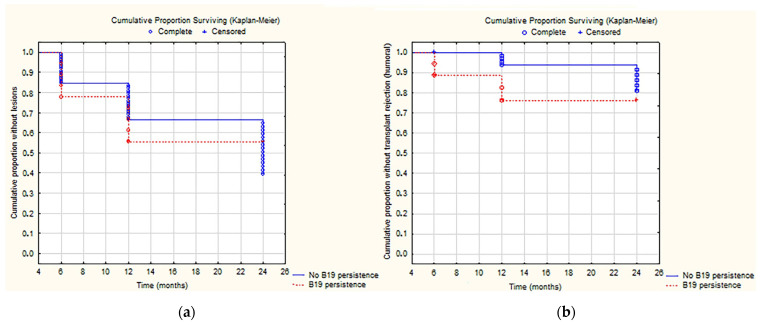
Analysis of cumulative incidence of acute or chronic allograft injury (**a**) and AMBR (**b**) and persistence of Parvovirus B19.

**Figure 2 ijms-24-09147-f002:**
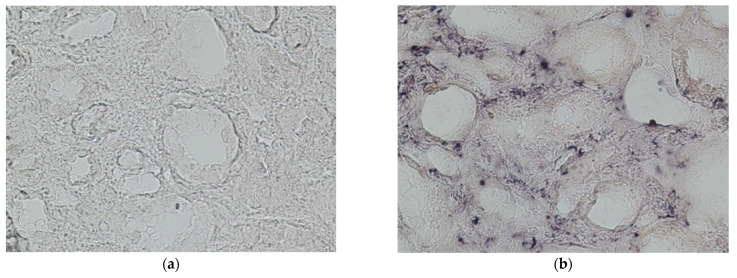
In situ hybridization for Parvovirus B19 in biopsy samples (40× magnification) with PCR positivity on tissue for Parvovirus B19 and TCMR (**a**) or AMBR (**b**).

**Table 1 ijms-24-09147-t001:** Population characteristics.

Patients enrolled	106
Age (years)	11 (r. 5–16)
Weight kg	26.9 ± 20.72 (r. 15.2–46.9)
Sex	
Male	67 (63.2%)
Female	39 (36.8%)
Donor	
Living	30 (28.3%)
Not living	76 (71.7%)
Kidney Transplant	
First	87
Second	18
D/R weight ratio kg	1.7 (r. 1.0–3.4)
(D/R) Mismatch HLA	3 (r. 3–4)

**Table 2 ijms-24-09147-t002:** Distribution of the results of protocol biopsies.

BIOPSIES	6 Months	12 Months	24 Months	Total
Banff1	72	54	27	153
ABMR	8	6	2	16
TCMR	10	10	16	36
IF/TA	2	2	5	9
Other diagnosis	0	1	3	4
Total	92	73	53	218

**Table 3 ijms-24-09147-t003:** Association between systemic viral infection and histological findings.

Detection of Viral DNA on Blood	Biopsies Results		*p*-Value
	6 Months of Follow-Up		
	Banff1	TCMR–ABMR–IF/TA	
Total	72 (78%)	20 (22%)	
Any virus	23 (32%)	1 (5%)	0.02
CMV	3 (4%)	0 (0%)	1
EBV	8 (11%)	1 (5%)	0.7
BKV	11 (15%)	0 (0%)	0.1
Parvovirus B19	3 (4%)	0 (0%)	1
	12 Months of Follow-Up		
Total	54 (75%)	18 (25%)	
Any virus	14 (26%)	5 (28%)	1
CMV	1 (2%)	0 (0%)	1
EBV	9 (17%)	2 (11%)	0.7
BKV	7 (13%)	3 (17%)	0.7
Parvovirus B19	1 (2%)	0 (0%)	1
	24 Months of Follow-Up		
Total	27 (54%)	23 (46%)	
Any virus	4 (15%)	9 (39%)	0.6
CMV	0 (0%)	0 (0%)	1
EBV	2 (7%)	5 (22%)	0.2
BKV	3 (11%)	5 (22%)	0.5
Parvovirus B19	0 (0%)	0 (0%)	1

**Table 4 ijms-24-09147-t004:** Association between intrarenal viral infection and histological findings.

Detection of Viral DNA on Bioptic Sample	Biopsies Results		*p*-Value
	6 Months of Follow-Up		
	Banff1	TCMR–ABMR–IF/TA	
Total	72 (78%)	20 (22%)	
Any virus	17 (24%)	5 (25%)	1
CMV	0 (0%)	0 (0%)	1
EBV	6 (8%)	1 (5%)	1
BKV	6 (8%)	1 (5%)	1
Parvovirus B19	17 (24%)	6 (30%)	0.6
	12 Months of Follow-Up		
Total	54 (75%)	18 (25%)	
Any virus	22 (41%)	10 (58%)	0.3
CMV	1 (1%)	0 (0%)	1
EBV	6 (11%)	3 (17%)	0.7
BKV	3 (6%)	2 (11%)	0.6
Parvovirus B19	14 (26%)	7 (39%)	0.4
	24 Months of Follow-Up		
Total	27 (54%)	23 (46%)	
Any virus	10 (37%)	11 (48%)	0.6
CMV	0 (0%)	0 (0%)	1
EBV	6 (22%)	7 (30%)	0.5
BKV	1 (4%)	4 (17%)	0.2
Parvovirus B19	4 (15%)	3 (13%)	1

**Table 5 ijms-24-09147-t005:** Association between systemic and intrarenal viral infection and type of rejection (ABMR vs. TCMR).

**Detection of Viral DNA on Blood**	**TCMR (n = 36)**	**AMBR (n = 16)**	** *p* ** **-Value**
Any virus	7 (19%)	4 (25%)	0.7
CMV	0 (0%)	0 (0%)	1
EBV	6 (17%)	2 (13%)	1
BKV	2 (5%)	3 (19%)	0.16
Parvovirus B19	0 (0%)	0 (0%)	1
**Detection of Viral DNA on Histological Sample**	**TCMR (n = 36)**	**AMBR (n = 16)**	** *p* ** **-Value**
Any virus	13 (36%)	11 (69%)	*p* < 0.05
CMV	0 (0%)	0 (0%)	1
EBV	6 (17%)	2 (13%)	1
BKV	3 (8%)	2 (13%)	0.6
Parvovirus B19	7 (19%)	8 (50%)	*p* < 0.05

## Data Availability

The data presented in this study are available on request from the corresponding author. The data are not publicly available due to privacy.

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
