# Peer review of "Pre-Existing Intrarenal Parvovirus B19 Infection May Relate to Antibody-Mediated Rejection in Pediatric Kidney Transplant Patients"

_ijms, 2023, doi:10.3390/ijms24119147_

Round 1

Reviewer 1 Report

These are interesting findings which lead to the hypothesis of a correlation of B19V infection of kidney transplants and antibody-mediated rejection.

There are some points which need to be improved:

I presume this is a retrospective study, but if I didn't miss it, it is not mentioned.

Figure 1 is quite blurred. Please improve readability.

There are many abbreviations used throughout the text. Some of them are introduced when first mentioned, but many not. Only in the materials and methods part they are all given, but this is at the end of the whole paper. Please give all abbreviations in one section or when first mentioned.

English language needs improvement concerning vocabulary and grammar (singular/plural).

Author Response

Dear reviewer,

I thank you for your kind suggestions. I will answer point-by-point.

  • I presume this is a retrospective study, but if I didn't miss it, it is not mentioned. --> I add it at the beginning of methods' chapter. (295)
  • Figure 1 is quite blurred. Please improve readability. --> I will upload the figure in more quality.
  • There are many abbreviations used throughout the text. Some of them are introduced when first mentioned, but many not. Only in the materials and methods part they are all given, but this is at the end of the whole paper. Please give all abbreviations in one section or when first mentioned. --> Thank you for your advice, I miss this point during the preparation of the manuscript and I am fixing it in the new version of it I will further upload.

I am also proceeding to improve English language.

Reviewer 2 Report

This study addressed the possible role of viral infections in transplant dysfunction and rejection, with a focus on the kidneys.

The manuscript has potential to be published, but it is necessary to present the information in a more logical way.

Minor points:

Table 2. Title is missed.

Figure 2. The legend is incomplete, should be expanded.

It is not clear how 218 histological samples were obtained from 106 patients enrolled in the study.

Major points:

There are at least two aspects missing:  no data on tissues at the moment of transplantation and the donor’s tissues status regarding the mentioned infections. Another issue is that there is no information on viral loads regarding each infection.

The title should be refined (it could be focused on all mentioned viruses). No well-defined aim of the study.

Different methods are mentioned (ELISA, IHC, LUMINEX) but related results are not shown in the article.

Please double-check the text for grammatical errors.

Author Response

Dear reviewer, 

Thank you for your suggestion. I think they will improve our manuscript. I will answer point-by-point.

Minor points:

Table 2. Title is missed. --> I will add it in the new version of manuscript. (102)

Figure 2. The legend is incomplete, should be expanded. --> I modify figure description (191-192). If you think it is not enough, can I ask you a kindly suggestion how to improve and to make clearer the description?

It is not clear how 218 histological samples were obtained from 106 patients enrolled in the study. -> not all patients underwent to kidney biopsy at 6, 12 and 24 months. In fact, some of them do not perform the kidney biopsy, due to many clinical conditions or lost of follow-up (i.e. adult transition). In fact the distribution of biopsies is listed in Table 2, but if you think it is not clear enough, I may introduce this concept or in methods (chapter 4.2) or in results (chapter 2.2).

Major points:

There are at least two aspects missing:  no data on tissues at the moment of transplantation and the donor’s tissues status regarding the mentioned infections. --> the only detectable virus on kidney at the moment of transplantation was Parvovirus B19, while the others were not detectable (138-140). If you think it makes the manuscript more complete I can add a new table with the results of infrarenal viral positivity on kidney at the moment of transplantation.

Another issue is that there is no information on viral loads regarding each infection. --> we consider the positivity/negativity of infection according to cut-off reported in methods' chapter (>1000 copies/ml of EBV-, CMV-, and BK virus-DNA for at least 3 of the 6 months before the biopsy). About the PCR on kidney biopsies only a positive/negative result was available. This is also true for the detection of Parvovirus B19 on blood sample. Due the relative small number of patient's positivity for the single viruses, it was not possible to stratify blood viral infections for the viral load, then we preferred to consider them only in a dihcotomic way (positive/negative).

The title should be refined (it could be focused on all mentioned viruses). No well-defined aim of the study. --> I modify the title according to your suggestion. I add a sentence with the aim of the study at the end of introduction (78-79).

Different methods are mentioned (ELISA, IHC, LUMINEX) but related results are not shown in the article. --> Forgive me, but I am not sure to understand what you want to suggest. Different tests were performed with different methods (i.e. DSA with Luminex), but in results we reported the positivity/negativity to the test according to cut off presented in methods' chapter. Moreover all tests performed on histological samples (chapter 4.2) were necessary to define Banff score. Finally, IHC results for Parvovirus B19 are shown in results (chapter 2.5), no other viruses were tested because not significative at previous investigations. 

Reviewer 3 Report

Review of Manuscript “Protocol Biopsies and Intrarenal Virus Infection: is there a Relationship between Parvovirus B19 and Humoral Rejection?” by Nicola..

The authors analyzed a total of 218 biopsies obtained from children undergoing renal transplantation at time points 6, 12 and 24 months post transplantation for viral infections involving CMV, EBV, VKV or Parvovirus B19 by RT-PCR analysis for viral transcripts. The manuscript is well written and the results are clearly presented.

For most, the authors found no significant associations between transplant rejection and systemic or renal presence of the different viruses. Only a weak possible relationship between the renal detection of parvovirus B19 (B19V) and antibody-mediated (ABMR) versus T-cell mediated (TCMR) rejection of the graft was observed. In the subsequent Kaplan Meier analysis, however, this was also not significant.

In view of this rather weak association between parvovirus B19 infection and AMBR, the part of the discussion addressing possible mechanism for a causal relationship between the infection and the development of AMBR seems to be quite speculative (lines 189 to 195). As the authors themselves state, especially the results of the analysis for active B19V replication in the kidney tissue should be performed on a larger number of samples. And finally one has to keep in mind that even a strong association would not necessarily point to a causal relationship.

In summary, the manuscript provides limited new findings regarding a possible relationship between viral infections and kidney transplant rejection. Additional major and minor points listed in detail below should also be addressed.           

Major Points:

1) Line 110 and following: From the lower prevalence of systemic viral infections in TCMR – ABMR  - IF/TA versus Banff1 6 months after transplantation one cannot conclude that viral infections may be protective against rejection. As already stated in the general remarks, an association may always be only a first hint towards a causal relationship.  

Minor points

1) Abstract and general: Abbreviations in the abstract such as ABMR and TCMR should be explained before first use. In general, explanations of abbreviations used are partially missing.

2) Line 22, typo: should read “it decreases“.

3) Line 62/63, form: “there are no specific therapies“ or “there is no specific therapy“.

4) A table 2 seems to be missing in the manuscript.

5) Line 150: Value for percentage of virus positive probes in TCMR in the text (34%) seems to differ from that in table 5 (36%).

Generally well written, some minor corrections as stated above.

Author Response

Dear reviewer,

I really appreciate your comment and I thank you for your suggestion to improve the manuscript. I will answer point-by-point.

Major Points:

1) Line 110 and following: From the lower prevalence of systemic viral infections in TCMR – ABMR  - IF/TA versus Banff1 6 months after transplantation one cannot conclude that viral infections may be protective against rejection. As already stated in the general remarks, an association may always be only a first hint towards a causal relationship.  --> You are completely right, so I modify the sentence to underline the association and not the relationship. I also modified the related sentence in discussion (289-290).

Minor points

1) Abstract and general: Abbreviations in the abstract such as ABMR and TCMR should be explained before first use. In general, explanations of abbreviations used are partially missing. --> I fix this problem according to other reviewer's suggestion, too. 

2) Line 22, typo: should read “it decreases“. --> modified

3) Line 62/63, form: “there are no specific therapies“ or “there is no specific therapy“. --> modified

4) A table 2 seems to be missing in the manuscript. --> I add the title of table 2

5) Line 150: Value for percentage of virus positive probes in TCMR in the text (34%) seems to differ from that in table 5 (36%). --> The correct percentage was 36%, I correct the value in the text.

According to your general comment, Do you suggest  to modify the discussion about possible mechanism for a causal relationship between the infection and the development of AMBR (lines 189 to 195)?